# "Ion-imprinting" strategy towards metal sulfide scavenger enables the highly selective capture of radiocesium

Jun-Hao Tang [1,2], Shao-Qing Jia[3], Jia-Ting Liu[1], Lu Yang[1], Hai-Yan Sun[1], Mei-Ling Feng [1,2,4] ✉ & Xiao-Ying Huang [1,2]

Highly selective capture of radiocesium is an urgent need for environmental radioactive contamination remediation and spent fuel disposal. Herein, a strategy is proposed for construction of "inorganic ion-imprinted adsorbents" with ion recognition-separation capabilities, and a metal sulfide $Cs_{2.33}Ga_{2.33}Sn_{1.67}S_8 \cdot H_2O$ (FJSM-CGTS) with "imprinting effect" on $Cs^+$ is prepared. We show that the $K^+$ activation product of FJSM-CGTS, $Cs_{0.51}K_{1.82}Ga_{2.33}Sn_{1.67}S_8 \cdot H_2O$ (FJMS-KCGTS), can reach adsorption equilibrium for $Cs^+$ within 5 min, with a maximum adsorption capacity of 246.65 mg·g$^{-1}$. FJMS-KCGTS overcomes the hindrance of $Cs^+$ adsorption by competing ions and realizes highly selective capture of $Cs^+$ in complex environments. It shows successful cleanup for actual $^{137}Cs$-liquid-wastes generated during industrial production with removal rates of over 99%. Ion-exchange column filled with FJMS-KCGTS can efficiently treat 540 mL $Cs^+$-containing solutions (31.995 mg·L$^{-1}$) and generates only 0.12 mL of solid waste, which enables waste solution volume reduction. Single-crystal structural analysis and density functional theory calculations are used to visualize the "ion-imprinting" process and confirm that the "imprinting effect" originates from the spatially confined effect of the framework. This work clearly reveals radiocesium capture mechanism and structure-function relationships that could inspire the development of efficient inorganic adsorbents for selective recognition and separation of key radionuclides.

$^{137}Cs$ ($t_{1/2}$ ~ 30.17 years) is one of the major sources of $\gamma$ radioactivity in spent fuel[1]. $^{137}Cs^+$ ions are highly soluble and migrate easily in the environment, and thus can be hazardous to human health and entire ecosystem[2,3]. For example, large quantities of $^{137}Cs^+$ ions have been released into the environment during the Fukushima nuclear accident, which has resulted in total radioactivity levels in a variety of organisms above the limit value (100 Bq kg$^{-1}$) after many years[4–6]. $^{137}Cs^+$ waste liquids can be generated during nuclear accidents, spent fuel disposal,

isotope source production, and other processes, in which there are usually a large number of non-radioactive ions such as $K^+$, $Na^+$, $Ca^{2+}$, and $Mg^{2+}$, as well as fission products such as $Sr^{2+}$ and lanthanide ions[2,7,8]. The complex environment and composition pose a great challenge to the selective separation of $Cs^+$ from various types of radioactive liquid waste. Ion exchange is considered to be an ideal method[9,10]. However, the anionic frameworks of traditional cation exchangers (e.g., ion-exchange resins, zeolites, clays, titanosilicates,

[1]State Key Laboratory of Structural Chemistry, Fujian Institute of Research on the Structure of Matter, Chinese Academy of Sciences, Fuzhou 350002 Fujian, PR China. [2]University of Chinese Academy of Sciences, 100049 Beijing, PR China. [3]HTA Co., Ltd., 102413 Beijing, PR China. [4]Fujian Province Joint Innovation Key Laboratory of Fuel and Materials in Clean Nuclear Energy System, Fujian Institute of Research on the Structure of Matter, Chinese Academy of Sciences, Fuzhou 350002, PR China. ✉e-mail: fml@fjirsm.ac.cn

and metal-organic frameworks) have strong Coulombic interactions with high-valency metal ions, and thus their $Cs^+$ capture properties are readily attenuated[7,9–13]. In addition, $K^+$ (212 pm) and $Na^+$ (228 pm) ions with similar hydration radii to $Cs^+$ (219 pm) also significantly affect the $Cs^+$ capture[7,12–14]. Metal sulfides are a new class of inorganic ion exchange materials developed in recent years[9,15–21]. The strong affinity of Lewis soft base $S^{2-}$ sites for Lewis soft acid $Cs^+$ ions and the flexible sulfide framework make them exhibit excellent removal ability for $Cs^+$ ions[9]. Recently, we have achieved the selective capture of $Cs^+$ in $HNO_3$ solutions containing excess interfering ions by a highly stable layered metal sulfide (InSnS-1) and confirmed that the selectivity originated from the $H^+$ inhibition mechanism and the $Cs^+ \cdots S^{2-}$ strong interaction[19]. Under neutral conditions, a very small number of three-dimensional microporous metal sulfide ion exchange materials with suitable pore sizes show good selectivity for $Cs^+$ capture, but the pore size limitation also makes these materials inefficient for $Cs^+$ removal, which is manifested by limited adsorption capacity and slow adsorption rate[9,22–25]. Although the adjustable interlayer spacing and readily exchangeable interlayer ions of layered sulfides allow them to capture $Cs^+$ efficiently, their $Cs^+$ capturing performance is greatly weakened by competing ions[9,26]. Therefore, it is still eager to develop the effective construction strategy for cesium scavengers to achieve efficient and highly selective capture of $Cs^+$ ions.

Molecular/ion imprinting is an emerging technology that has been widely studied and applied in the field of water treatment[27–29]. Imprinted adsorbents synthesized using customized templates (functional groups or functional monomers) have specific recognition-adsorption capabilities, which have been used for the removal of heavy metal ions and radioactive ions[28,30,31]. However, ion-imprinted polymers for selective $Cs^+$ capture are uncommon due to the limitation of functional monomer species[30,31]. Herein, we propose a strategy to use ion-imprinting technology for the preparation of inorganic materials. The imprinting effect of functional monomers on $Cs^+$ was simulated by taking advantage of the relatively robust framework structure of inorganic materials and the spatially confined effect of affinity sites on $Cs^+$. The specific synthesis strategy is as follows. Firstly, non-radioactive $Cs^+$ ions are used to participate in material synthesis to construct frameworks with strong coordination ability for target nuclides ions ($Cs^+$). Then, the non-radioactive $Cs^+$ ions are removed from the material (e.g., activation by high concentration of $K^+$ ions), leaving sites with a strong affinity for the target nuclides ions ($Cs^+$) as well as a suitable space. This will allow the anionic framework of material to specifically recognize the target nuclides ions ($Cs^+$). In this work, the ion imprinting technique is successfully promoted to the preparation of inorganic adsorbents. The current strategy will combine the high selectivity advantage of imprinted adsorbents with the excellent irradiation resistance, environmental compatibility, and adsorption efficiency of inorganic adsorbents. Thus, the prepared "inorganic ion-imprinted adsorbents" will be more advantageous to radioactive waste treatment which generally faces the harsh environment. Based on this strategy, the selectivity of $Cs^+$ ion-imprinted metal sulfide scavenger for $Cs^+$ will further be enhanced due to the affinity of soft basic $S^{2-}$ sites for $Cs^+$.

Guided by the aforementioned strategy, we have successfully constructed a layered gallium thiostannate $Cs_{2.33}Ga_{2.33}Sn_{1.67}S_8 \cdot H_2O$ (FJSM-CGTS) that exhibits the imprinting effect on $Cs^+$. Its $K^+$ activation product, $Cs_{0.51}K_{1.82}Ga_{2.33}Sn_{1.67}S_8 \cdot H_2O$ (FJMS-KCGTS), exhibits excellent selectivity for $Cs^+$ removal. In $Cs^+$ solutions with excess competing ions ($K^+$, $Ca^{2+}$, $Na^+$, $Mg^{2+}$, $Sr^{2+}$, and $Eu^{3+}$) or in environmental water samples, it can effectively remove low-concentration $Cs^+$ ions. It is recyclable and can be used as a stationary phase in ion exchange columns for fast and easy treatment of $Cs^+$-containing solutions to achieve waste volume minimization. Attractively, FJSM-KCGTS demonstrates excellent treatment capabilities for actual $^{137}$Cs-liquid-waste generated during industrial production. Moreover, the "ion

imprinting" process was visually revealed by the single-crystal structure analyses. The selective adsorption mechanism and structure-function relationship were clarified by structural comparative analysis and density functional theory (DFT) theoretical calculations from perspectives of structure and energy. The validity of the synthetic strategy for constructing "inorganic ion-imprinted adsorbents" was confirmed. It is unprecedented to use ion imprinting method for the highly selective capture of $Cs^+$ by inorganic materials. This work provides inspiration for the development of inorganic ion-imprinted scavengers with high selectivity for target radionuclides removal.

## Results

### Synthesis and characterization

The light-pink plate-like crystals of FJSM-CGTS ($Cs_{2.33}Ga_{2.33}Sn_{1.67}S_8 \cdot H_2O$) can be easily synthesized in one step by solvothermal method. KCl solution (2 mol $L^{-1}$) was used to activate FJSM-CGTS. The vast majority of $Cs^+$ in the interlayer of FJSM-CGTS were replaced by $K^+$ to produce FJSM-KCGTS ($Cs_{0.51}K_{1.82}Ga_{2.33}Sn_{1.67}S_8 \cdot H_2O$, Supplementary Method 2.1). $K^+$ in FJSM-KCGTS was in turn exchanged for $Cs^+$, yielding the adsorption product, FJSM-KCGTS-Cs ($Cs_{2.12}K_{0.21}Ga_{2.33}Sn_{1.67}S_8 \cdot H_2O$). Molecular formulae were determined from single-crystal structure analysis (Supplementary Tables 1–5) and energy dispersive spectroscopy (EDS, Supplementary Fig. 1), thermogravimetric analysis (Supplementary Fig. 2) and inductively coupled plasma-optical emission spectroscopy (ICP-OES) test. Powder X-ray diffraction (PXRD) patterns are in agreement with three corresponding simulated PXRD pattern calculated from single-crystal X-ray data (Supplementary Fig. 3). Single-crystal X-ray diffraction, EDS, elemental distribution maps and X-ray photoelectron spectroscopy (XPS, Supplementary Method 2.2) tests identify successful exchange between $K^+$ and $Cs^+$, confirming the "ion imprinting" process (Supplementary Figs. 3–5). Scanning electron microscope (SEM) images show that the ion exchange process of $K^+$ and $Cs^+$ had no significant effect on the morphology of materials (Supplementary Fig. 6).

FJSM-CGTS crystallizes in the $Pmc2_1$ space group. T2-$[Ga_{2.33}Sn_{1.67}S_{10}]^{6.33-}$ clusters constructed from corner-sharing $[(Sn/Ga)S_4]$ tetrahedra act as the secondary building blocks for the anionic layers of $[Ga_{2.33}Sn_{1.67}S_8]^{2.33n-}$ in FJSM-CGTS (Fig. 1a). Each T2 cluster building block shares its four corners ($\mu_2$-$S^{2-}$) with four neighboring T2 clusters, forming a wavy $[Ga_{2.33}Sn_{1.67}S_8]_n^{2.33n-}$ layer (Fig. 1a). $Cs^+$ are distributed in an unorganized manner in the interlayer spaces (Fig. 1a). The close proximity between Cs2 and Cs2B leads to the inability of both positions to be occupied by $Cs^+$ at the same time, and similarly for Cs3 and Cs3B. By single-crystal structure analysis, both FJSM-KCGTS and FJSM-KCGTS-Cs maintain the layer structure of the parent (Supplementary Fig. 7). The schematic diagram of the "ion imprinting" process is shown in Fig. 1b. In FJSM-KCGTS, $Cs^+$ in the original Cs1 and Cs3/Cs3B positions were completely replaced by $K^+$ and $Cs^+$ in the Cs2/Cs2B positions were partially replaced by $K^+$. In FJSM-KCGTS-Cs, $K^+$ in the original Cs1 and Cs3/Cs3B positions were completely replaced by Cs and $K^+$ in the Cs2/Cs2B positions were partially replaced by $Cs^+$. This may be due to the fact that Cs1 and Cs3/Cs3B are located in the interlayer spaces, and they are coordinated with seven or eight sulfurs from the different layers, respectively, whereas Cs2/Cs2B situate within the layer and are coordinated by eight sulfurs from the same layer (Fig. 1c and Supplementary Fig. 8). Therefore, Cs2/Cs2B are more difficult for ion exchange compared with Cs1 and Cs3/Cs3B resulting in the partial replacement of $K^+$ in Cs2/Cs2B positions. *ORTEP*-style illustrations of FJSM-CGTS, FJSM-KCGTS and FJSM-KCGTS-Cs are shown in Supplementary Fig. 9.

### Efficient extraction of $Cs^+$

FSJM-KCGTS obtained by $K^+$ activation of FJSM-CGTS (Supplementary Method 2.1 and Supplementary Fig. 10) was used for $Cs^+$ capture performance studies. The $Cs^+$ adsorption of FJSM-KCGTS can reach

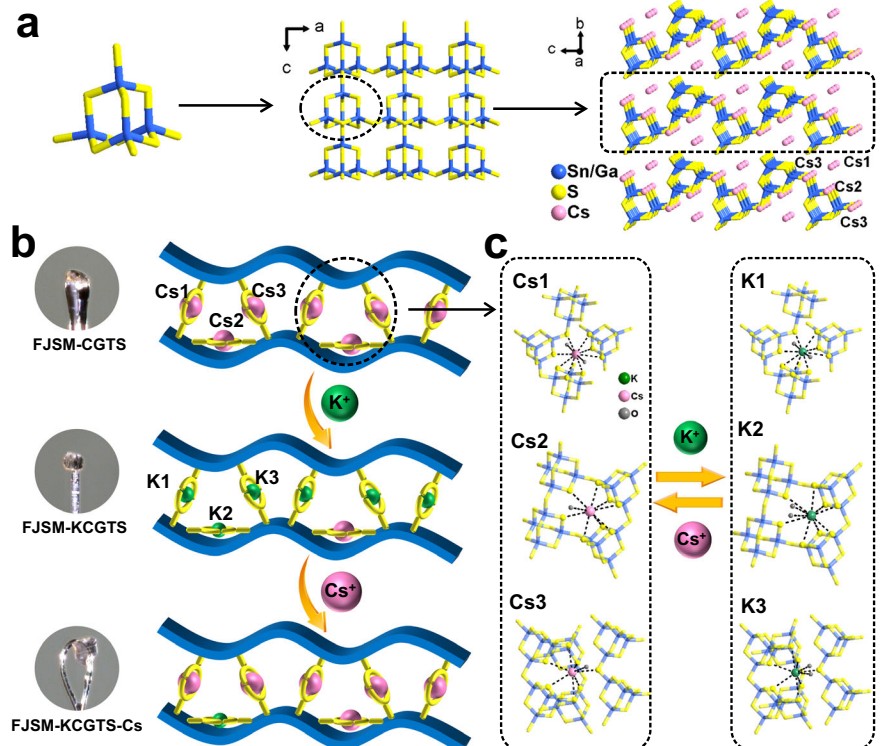

**Fig. 1 | Schematic diagram of the structure of FJSM-CGTS and the "ion imprinting" process. a** A T2-$[Ga_{2.33}Sn_{1.67}S_{10}]^{6.33-}$ cluster, an anionic layer of $[Ga_{2.33}Sn_{1.67}S_8]_n^{2.33n-}$ and view of layers stacking in FJSM-CGTS along $a$ axis. Cs2B, Cs3B, O, and H are ignored for clarity. **b** Schematic diagram of the "ion imprinting" process for the selective capture of $Cs^+$ by the current layered metal sulfide and the single-crystals photographs of FJSM-CGTS, FJSM-KCGTS and FJSM-KCGTS-Cs. The blue wavy layer and the yellow hoop are abstracted representations of the $[Ga_{2.33}Sn_{1.67}S_8]_n^{2.33n-}$ anionic layer. **c** Coordination patterns of Cs1, Cs2, Cs3 in FJSM-CGTS and K1, K2, K3 in FJSM-KCGTS.

equilibrium within 5 min with the removal rate ($R$, Eq. (S1)) exceeding 99% (Fig. 2a), and the kinetic data are highly fitted with the pseudo-first-order kinetic model (Supplementary Fig. 11 and Supplementary Table 6). The $Cs^+$ concentration is dramatically reduced to a few tens of micrograms per liter, a reduction in concentration of two orders of magnitude. The $Cs^+$ adsorption isotherm data of FJSM-KCGTS at room temperature (RT) exhibits well-fitting to the Langmuir isotherm model with $R^2$ of 0.99 and the maximum exchange capacity ($q_m$) of 246.65 mg g$^{-1}$ (Fig. 2b and Supplementary Table 7). The theoretical exchange capacity of FJSM-KCGTS for $Cs^+$ was 287.52 mg g$^{-1}$, calculated from the chemical formulae of FJSM-KCGTS and FJSM-KCGTS-Cs. The experimental values were slightly lower than the theoretical value which may be attributed to the fact that the exchanged $K^+$ would compete with $Cs^+$. $q_m^{Cs}$ of FJSM-KCGTS is higher than the reported $Cs^+$ ion-imprinted adsorbents, including Cs(I)-IIP2 (54.54 mg g$^{-1}$)[32], cesium ion-imprinted polymer based on dibenzo-24-crown-8 ether nanoparticles (50 mg g$^{-1}$)[33], and Cs(I) ion-imprinted polymer based on sodium trititanate whisker and chitosan (32.88 mg g$^{-1}$)[34]. It is also more than many common $Cs^+$ adsorbents including zeolites (e.g., Turkish samples: 89.18 mg g$^{-1}$)[35], carbon-based (e.g., GO: 76.9 mg g$^{-1}$) materials[36], metal sulfides (e.g., KMS-1: 226 mg g$^{-1}$)[37], and the commercially available AMP-PAN (81 mg g$^{-1}$) marketed by UOP as IONSIV IE-91[38].

FJSM-KCGTS is also acid-base resistant, irradiation resistant and recyclable. The distribution coefficients ($K_d$) of FJSM-KCGTS for $Cs^+$ exceeds 10³ mL g$^{-1}$ over a wide pH range from 1.99 to 11.84 (Supplementary Fig. 12a). In pH range of 3.94–10.0, $K_d^{Cs}$ exceed 10⁵ mL g$^{-1}$ with $R^{Cs}$ above 99%. When Cs/Sr coexists, the $Cs^+$ capture of FJSM-KCGTS is almost unaffected. At pH = 3.23–10.51, $R^{Cs}$ of FJSM-KCGTS exceeds 96% with $K_d^{Cs} > 10^4$ mL g$^{-1}$, which confirms the strong affinity of FJSM-KCGTS for $Cs^+$ (Fig. 2c and Supplementary Fig. 12b). Such excellent $Cs^+$

adsorption property of FJSM-KCGTS can be retained after irradiation with 100 and 200 kGy $\gamma$-rays, with $R^{Cs} > 99.5\%$ and $K_d^{Cs} > 10^5$ mL g$^{-1}$ (Fig. 2d). In addition, the $Cs^+$ adsorbed material can be regenerated by elution with 1 mol L$^{-1}$ NH$_4$Cl solution (Supplementary Fig. 13), and the $R^{Cs}$ can be maintained above 98.97% after three adsorption-elution cycles (Supplementary Method 2.3 and Supplementary Fig. 14). PXRD confirms that the framework of FJSM-KCGTS can be maintained within pH = 2.14–11.48 and after three adsorption-elution cycles, and its parent structure can be retained after $\gamma$-ray irradiation (Supplementary Fig. 15a–c). Low leaching rates of Ga$^{3+}$ (0.026–3.81%) can be observed over a wide pH range of pH = 3.23–10.51 (Supplementary Fig. 15d). The excellent stability of FJSM-KCGTS makes it a potential radionuclide scavenger.

## Selective capture of $Cs^+$

The effect of competing ions (Na$^+$, K$^+$, Ca$^{2+}$, Mg$^{2+}$, Sr$^{2+}$, and Eu$^{3+}$) on the selective capture behavior of $Cs^+$ ions by FJSM-KCGTS was systematically examined. $K_d^{Cs}$ of FJSM-KCGTS can be maintained above 10⁴ mL g$^{-1}$ with $R^{Cs} > 91\%$ when Na/Cs molar ratios are in the range of 27.1–1246. Even when Na/Cs molar ratio reaches $1.13 \times 10^4$, $K_d^{Cs}$ is still above 10³ mL g$^{-1}$, with $R^{Cs}$ of 58.80% (Fig. 3a). Since the material contains $K^+$, the excess $K^+$ in solution is bound to greatly affect the $Cs^+$ exchange compared to Na$^+$ ions (Supplementary Fig. 16). Even so, when K/Cs molar ratio is 408, $K_d^{Cs}$ can attain $1.71 \times 10^3$ mL g$^{-1}$ with $R^{Cs}$ of 63.11%. When $Cs^+$ coexists with the high-valency competing ion $M^{n+}$ ($M^{n+} = Ca^{2+}, Mg^{2+}, Sr^{2+}$, and $Eu^{3+}$), highly selective capture of $Cs^+$ by FJSM-KCGTS can also be observed (Fig. 3b–e and Supplementary Fig. 17). Under the low molar ratio of $M$/Cs, $R^{Cs}$ are more than 97.7% and $K_d^{Cs}$ are higher than 10⁴ mL g$^{-1}$. At this time, $R^M$ can also reach more than 68%. When the $M$/Cs molar ratio is increased by two orders of magnitude, $R^{Cs}$ decrease slightly, but still reach more than 73% with $K_d > 10^3$ mL g$^{-1}$.

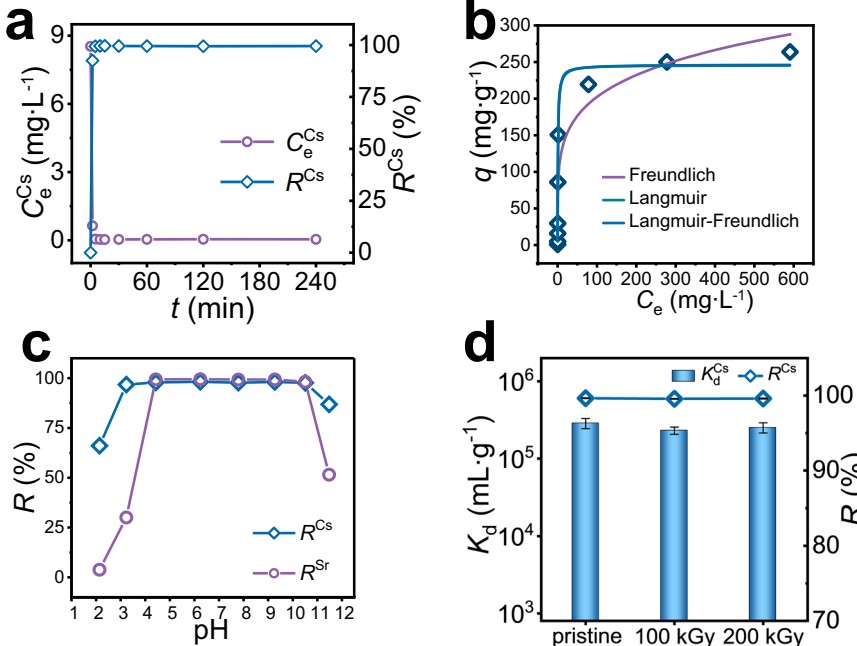

**Fig. 2 | Removal performance of FJSM-KCGTS for Cs⁺. a** Kinetics of Cs⁺ removal by FJSM-KCGTS plotted as Cs⁺ concentration and $R^{Cs}$ vs. time $t$ (min), respectively. **b** Equilibrium data for Cs⁺ removal by FJSM-KCGTS fitted with the Langmuir, Freundlich and Langmuir-Freundlich isotherm models. **c** $R^{Cs}$ and $R^{Sr}$ values of FJSM- KCGTS in coexisting Cs/Sr solutions with various initial pH values. **d** $K_d{}^{Cs}$ and $R^{Cs}$ values of FJSM-KCGTS samples before and after irradiation. Error bars present the standard deviation of the mean of three experiments. Source data are provided as a Source Data file.

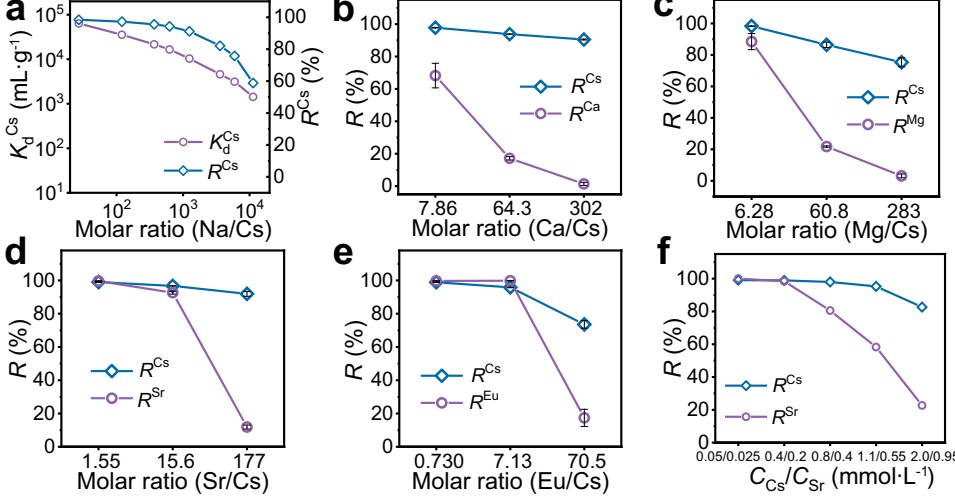

**Fig. 3 | Selective removal of Cs⁺ by FJSM-KCGTS in the presence of competing ions.** The selective capture of Cs⁺ by FJSM-KCGTS in the presence of competing ions. $K_d{}^{Cs}$ and $R^{Cs}$ values of FJSM-KCGTS in neutral solutions with different (**a**) Na/Cs molar ratios. $R$ of Cs⁺ and $M^{n+}$ ($M^{n+}$ = Ca²⁺, Mg²⁺, Sr²⁺, and Eu³⁺) ions removed by FJSM-KCGTS in neutral solutions with different (**b**) Ca/Cs, (**c**) Mg/Cs, (**d**) Sr/Cs, and (**e**) Eu/Cs molar ratios. **f** $R^{Cs}$ and $R^{Sr}$ of FJSM-KCGTS in neutral Sr/Cs solutions with equipotent charge concentration ($C_0{}^{Cs} = 2C_0{}^{Sr}$). Error bars present the standard deviation of the mean of three experiments. Source data are provided as a Source Data file.

The decrease of $R^{Cs}$ is relatively big in the coexisting Eu/Cs solution, from 98.90% to 73.55%, and the smallest decrease in $R^{Cs}$ is observed in the coexisting Ca/Cs solution, from 97.79% to 90.51%. By contrast, $R^M$ decreases dramatically, that is, $R^{Ca}$, $R^{Mg}$, $R^{Sr}$, and $R^{Eu}$ decreased to 1.39%, 3.12%, 11.86%, and 17.39%, respectively. The above results show that FJSM-KCGTS had removal ability for both Cs⁺ and $M^{n+}$ under the low concentrations of both Cs⁺ and $M^{n+}$. However, when the $M^{n+}$ concentration is substantially increased, the adsorption performance of FJSM-KCGTS for Cs⁺ is maintained at a high level, while the adsorption performance for $M^{n+}$ is significantly reduced. That is to say, FJSM-

KCGTS can still selectively capture low concentrations of Cs⁺ ions in the presence of a large number of competing ions.

We believe that the adsorption performance of FJSM-KCGTS for both Cs⁺ and $M^{n+}$ ions in low concentration solutions may originate from the redundancy of ion exchange active sites. Therefore, taking the coexisting Cs/Sr system as a representative, we further investigated the adsorption behaviors of FJSM-KCGTS under coexisting Cs/Sr with both of equimolar concentrations or equipotent charge concentrations. The same results are found under both conditions (Fig. 3f and Supplementary Fig. 18). Under low Cs⁺ and Sr²⁺ ion concentrations,

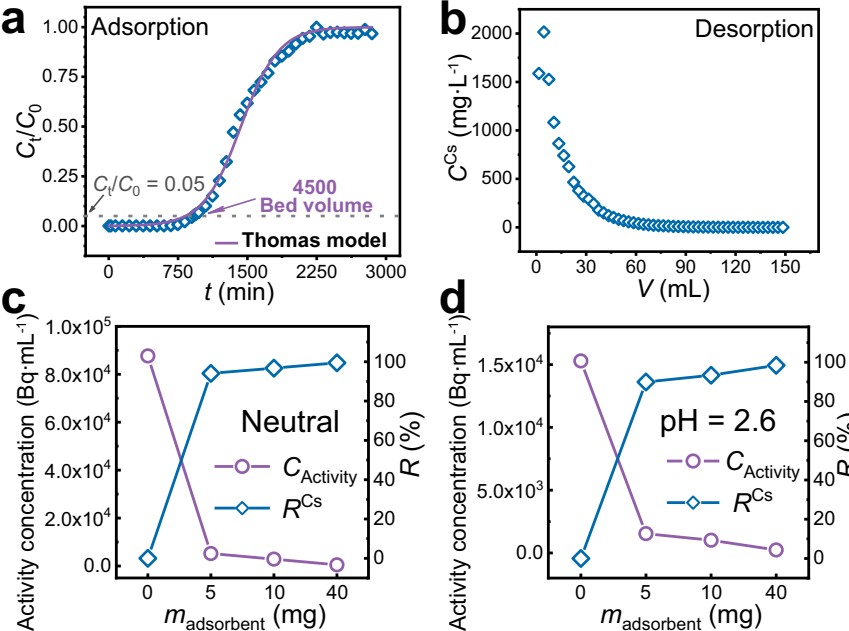

**Fig. 4 | Exploration of the practical application potential of FJSM-KCGTS for Cs⁺ capture. a** The Cs⁺ capture breakthrough curve and (**b**) elution curve of FJSM-KCGTS packed column. Treatment capacity of FJSM-KCGTS for (**c**) neutral and (**d**) acidic (pH = 2.6) $^{137}$Cs liquid-waste plotted as the activity concentration and the $R$ of $^{137}$Cs⁺ vs. the mass of adsorbent (mg), respectively. Source data are provided as a Source Data file.

FJSM-KCGTS shows good removal performance for both Cs⁺ and Sr²⁺ ions, with $K_d^{Cs} > 10^5$ mL g⁻¹, $K_d^{Sr} > 10^6$ mL g⁻¹, $R^{Cs} > 99\%$, and $R^{Sr} > 99.9\%$. With the increase of both concentrations, the capture performance of FJSM-KCGTS for Sr²⁺ decreases significantly, while the capture performance for Cs⁺ is maintained. Specifically, in coexisting Cs/Sr solutions with equimolar concentrations or equipotent charge concentrations, $R^{Cs}$ can still reach 82.65% and 75.74% whereas $R^{Sr}$ decreases to 22.70% and 16.20%, respectively. This is attributed to the fact that under low concentrations of Cs/Sr ions, FJSM-KCGTS has an excess of ion exchange active sites, thus exhibiting removal of both Cs⁺ and Sr²⁺. By contrast, the limited adsorption sites of FJSM-KCGTS can not accommodate all Cs⁺ and Sr²⁺ ions under high concentration of Cs⁺/Sr²⁺ ions. Note that the removal performance of FJSM-KCGTS for Cs⁺ still remain at a high level while the removal performance for Sr²⁺ decreases significantly, indicating that FJSM-KCGTS has the ability to preferentially selectively capture Cs⁺. This excellent selectivity enable FJSM-KCGTS to achieve effective removal of Cs⁺ from lake, river and seawater samples with added Cs⁺. Especially in complex seawater samples, it is rare that $q_m^{Cs}$ can still reach 206.44 mg g⁻¹, proving that FJSM-KCGTS has the ultra-high selectivity for Cs⁺ removal in highly saline environments (Supplementary Method 2.4, Supplementary Fig. 19 and Supplementary Table 8).

Therefore, the selective Cs⁺ capture ability of FJSM-KCGTS is little weakened by the increased concentration of competing ions. It is noteworthy that this phenomenon is quite different from the reported results. The Cs⁺ adsorption properties of most Cs⁺ scavengers such as MOFs, clay minerals, and even some metal sulfide materials are significantly weakened by competing ions, especially high-valency ions[13,37,39-41]. Some metal sulfide ion-exchange materials containing protonated organic amines can also capture Cs⁺ after K⁺ activation, but their capturing performance is also very susceptible to competing ions[40,42,43]. A very small number of three-dimensional materials with suitable pore size have good selectivity for Cs⁺ capture[9,22-25]. However, we found that the presence of competing Sr²⁺ and Eu³⁺ ions is also detrimental to their capture efficiency for Cs⁺ (Supplementary Method 2.5 and Supplementary Figs. 20 and 21). The selective Cs⁺ capture performance of FJSM-KCGTS synthesized under the guidance of our strategy is at the forefront of reported materials, with high adsorption capacity, short adsorption equilibrium time and high selectivity. It suggests that the current "ion-imprinting effect" plays a key role in the Cs⁺ capture process of FJSM-KCGTS. The design and synthesis strategy of the current material should be effective in significantly enhancing the selectivity for Cs⁺ removal in neutral high-salinity solutions.

## Column experiments
FJSM-KCGTS was packed into the ion exchange column as stationary phase to realize the adsorption and elution process of Cs⁺ conveniently and efficiently (Supplementary Fig. 22). The breakthrough curve of Cs⁺ ion adsorption conforms to the Thomas model ($R^2 > 0.99$) with $q_m^{Cs}$ of 288.94 mg g⁻¹, and a treatment volume of about 4500 bed volumes when the breakthrough point ($C_t/C_0 = 0.05$) is reached (Fig. 4a and Supplementary Table 9). In other words, about 540 mL of Cs⁺ solution ($C_0 = 31.995$ mg L⁻¹) can be effectively treated to produce only one bed volume (0.12 mL) of solid waste, resulting in a three-order-of-magnitude reduction in waste volume. Note that unlike isothermal adsorption experiments, the exchanged K⁺ ions are eluted out of the adsorption column in the column experiments and do not affect the adsorption of Cs⁺. Therefore, the dynamic adsorption capacity is comparable to the theoretical value (287.52 mg g⁻¹). Such dynamic capture performance of Cs⁺ ions exceeds most reported Cs⁺ ion exchangers such as InSnS-1, MCC-g-AMP, M/SZMs, and PEI/ZnFC[19,44-46]. After reaching the saturation adsorption, the adsorbed Cs⁺ ions can be leached off with 1 mol L⁻¹ NH₄Cl solution (Fig. 4b). It is noteworthy that the Cs⁺ ions concentration in the first four eluent samples (totaling 12 mL) reaches more than 10³ mg L⁻¹. The first 16 eluent samples (totaling 12 mL) can elute 32.10 mg of Cs⁺ ions with elution rate ($R_E$) of 95.65% (Supplementary Method 2.5). Therefore, the ion exchange column filled with FJSM-KCGTS can quickly and effectively enrich and concentrate Cs⁺ ions. FJSM-KCGTS has the potential for engineering treatment of radiocesium-containing waste liquids to realize waste liquid volume reduction.

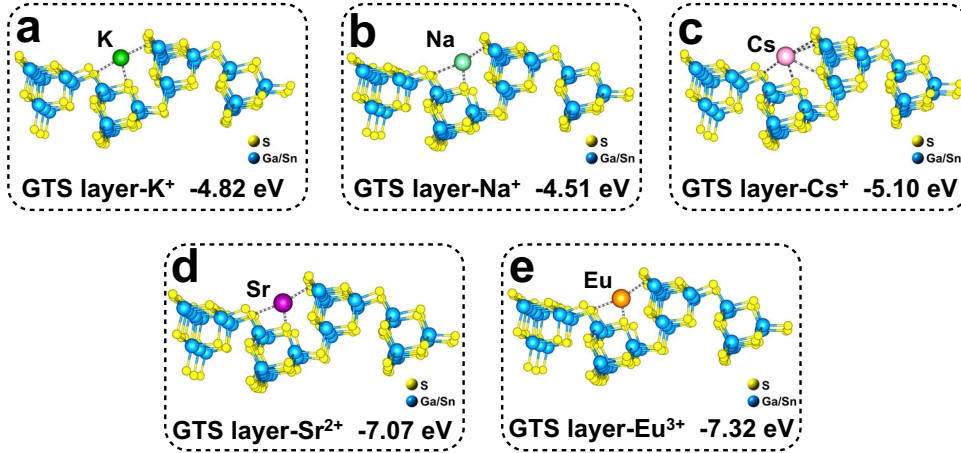

**Fig. 5 | Results of density functional theory calculations.** The calculation on binding energies of the $[Ga_{2.33}Sn_{1.67}S_8]_n^{2.33n-}$ layer (abbreviated as "GTS layer" in the figures) to (**a**) $K^+$, (**b**) $Na^+$, (**c**) $Cs^+$, (**d**) $Sr^{2+}$, and (**e**) $Eu^{3+}$ ions. The atomic coordinates of the optimized computational models are provided in Supplementary Data 1.

## Actual $^{137}Cs$-liquid-waste treatment

FJSM-KCGTS was used for the treatment of neutral and acidic (pH = 2.6) $^{137}Cs$-liquid-waste generated from the production of radioactive sources at CNNC HTA Co., ltd. Both solutions contained $^{60}Co^{2+}$ and $Na^+$ ions and so on in addition to $^{137}Cs^+$. The experimental results show that only 5 mg of FJSM-KCGTS can remove 94.09% and 89.93% of $^{137}Cs$ from 5 mL neutral and acidic $^{137}Cs$-liquid-waste, respectively (Fig. 4c, d). When 40 mg of FJSM-KCGTS is used (solid-liquid ratio $m/V = 8\,mg\,mL^{-1}$), $R^{Cs-137}$ reaches 99.45% and 98.42%, respectively, and the $^{137}Cs$ activity concentrations of both wastewaters are reduced by two orders of magnitude. Therefore, a small amount of FJSM-KCGTS can significantly reduce the radioactivity of wastewater, which contributes to the effective reduction of the volume of radioactive waste. FJSM-KCGTS shows excellent $^{137}Cs$ removal performance under actual working conditions, which highlights a $^{137}Cs$ scavenger with practical application potential.

## Density functional theory calculations

In order to further elucidate the mechanism of the highly selective capture of $Cs^+$ by FJSM-KCGTS, DFT calculations of the interaction of the $[Ga_{2.33}Sn_{1.67}S_8]_n^{2.33n-}$ layer with $M^{n+}$ ($M^{n+} = Cs^+$, $K^+$, $Na^+$, $Sr^{2+}$, and $Eu^{3+}$) ions were performed. DFT results show that the binding energies of $Cs^+$, $K^+$, $Na^+$, $Sr^{2+}$, and $Eu^{3+}$ ions to anionic layers are −5.10, −4.82, −4.51, −7.07, and −7.32 eV, respectively (Fig. 5). The binding energies of $Na^+$ and $K^+$ to anionic layers are lower than that of $Cs^+$, suggesting stronger interactions between anionic layers and $Cs^+$ ions. The binding energies of $Sr^{2+}$ and $Eu^{3+}$ ions to anionic layers are higher than that of $K^+$ ions, but each $Sr^{2+}$ and $Eu^{3+}$ ion need to displace two and three $K^+$ ions, respectively. Therefore, this process requires a higher energy than the replacement of $K^+$ ions with $Cs^+$ ions. DFT results confirm that FJSM-KCGTS possesses the stronger selectivity for $Cs^+$ than $Na^+$, $K^+$, $Sr^{2+}$, and $Eu^{3+}$ from view of energy, which are consistent with the phenomena observed in systematic selectivity experiments.

## Capture mechanism and structure-function relationship

Combined with the above experiments and characterization data, the selective capture ability of FJSM-KCGTS for $Cs^+$ is believed to originate from the synergistic action of the ion exchange and the imprinting effect. Single-crystal structure analyses clearly and visually demonstrate the whole process of "ion-imprinted" $Cs^+$ capture by FJSM-CGTS as well as the coordination pattern of Cs···S. FJSM-CGTS shows more outstanding $Cs^+$ selectivity than reported $Cs^+$ ion exchangers such as KMS-1[37], KMS-2[16], KTS-3[17], and FJSM-SnS[18]. Because the anionic layers of these reported $Cs^+$ ions scavengers are plate-like (Supplementary Fig. 23a), the entry of $Cs^+$ ions into the interlayer space causes a significant change in layer spacing (Supplementary Table 10). For example, KMS-1 and FJSM-SnS captured $Cs^+$ with an increase or decrease in layer spacing of 0.49 Å and 0.549 Å, respectively. The flexible and adjustable interlayer spacing characteristic of these ion exchangers makes them easily accommodate foreign ions, which may lead to their relatively poor selective capture of $Cs^+$. However, the $[Ga_{2.33}Sn_{1.67}S_8]_n^{2.33n-}$ anionic layers of FJSM-KCGTS are wavy-like (Supplementary Fig. 23b). During $K^+$ activation and $Cs^+$ adsorption, the layer spacing variation of FJSM-CGTS was less than 0.25 Å. This may be due to the fact that the wavy layers stacking pattern produces stronger interlayer interactions than planar layers stacking pattern, favoring the reduction of layer spacing variations. The relatively robust wavy-layer structure of FJSM-CGTS provides a suitable spatial and coordination environment for $Cs^+$, which can be regarded as "$Cs^+$ recognition sites". These sites are more suitable for accommodating and "anchoring" $Cs^+$ ions, even after the sites are occupied by $K^+$ ions. Therefore, we suggest that the wavy layered structure and the strong interaction of soft basic $S^{2-}$ sites with $Cs^+$ provide a spatially confined effect on $Cs^+$, contributing to the excellent selectivity of FJSM-KCGTS for $Cs^+$ capture. DFT results also confirm this point from an energetic viewpoint.

The unparalleled $Cs^+$ capture performance of FJSM-CGTS (FJSM-KCGTS) proves the effectiveness of the "inorganic ion-imprinted adsorbent" synthesis strategy. The advantages of ion exchangers and imprinted adsorbents were combined to make "inorganic ion-imprinted adsorbents" with high adsorption efficiency and outstanding selectivity. By utilizing the "imprinting effect" generated by the spatially confined effect, highly selective recognition and capture of target ions can be achieved. Notably, the "imprinting effect" of the current material originating from a specific structure of inorganic material is expected to reduce the constraints imposed by the type of functional groups or functional monomers on the development of imprinted polymers because of the variety, tunability, and designability of the frameworks of inorganic materials.

## Discussion

We propose a strategy to construct the "inorganic ion-imprinted adsorbent" that utilizes the spatially confined effect from the robust framework of inorganic materials to achieve "imprinted adsorption" of target ions. Under the synergistic action of ion exchange and "imprinting effect", the inorganic layered metal sulfide FJSM-CGTS (FJSM-KCGTS) prepared based on this strategy overcomes the defect that the performance of traditional $Cs^+$ ion exchangers is greatly weakened by competing ions, especially high-valency ions, and it can selectively recognize and capture $Cs^+$. It can be used as a stationary

phase for ion-exchange columns to realize the volume reduction and concentration of $Cs^+$-containing solutions quickly and conveniently. Importantly, FJSM-KCGTS demonstrates excellent treatment capability for actual $^{137}Cs$-liquid-waste generated during industrial production. Single-crystal structure analysis "visualizes" the whole process of "ion-imprinted" adsorption for $Cs^+$, and theoretical calculations reveal that the strong binding force of the anionic sulfide framework to $Cs^+$ is the source of the high selectivity of FJSM-KCGTS. The excellent $Cs^+$ capture performance of FJSM-KCGTS proves the effectiveness of our synthesis strategy. The high selectivity of the imprinted adsorbent and the stability, high adsorption efficiency and good environmental compatibility of the inorganic adsorbent material can be combined through this strategy to prepare a new "inorganic ion-imprinted adsorbent" that is more suitable for the treatment of radioactive waste liquids. This study pioneers the preparation of inorganic materials with ion-imprinting functionality. The clear illumination on the selective adsorption mechanism and structure-function relationship provides the direction for developing efficient inorganic adsorbents with the ability of specifically recognition-separation for key radionuclides.

## Methods

### Starting materials

$Cs_2CO_3$ (99.90%, Adamas), S (CP, Sichuan Kelong Chemical Co., Ltd.), Sn (99%, Damas-beta), $Ga(NO_3)_3 \cdot xH_2O$ (99.99%, Shanghai Longjin Metal Materials Co., Ltd.), $N_2H_4 \cdot H_2O$ (>98.0%, Aladdin), CsCl (99.99%, Shanghai Longjin Metal Materials Co., Ltd.), $SrCl_2 \cdot 6H_2O$ (AR, Tianjin Guangfu Reagent Co., Ltd.), $EuCl_3 \cdot 6H_2O$ (99.99%, Ruike Rare Earth Metallurgy and Functional Materials National Engineering Research Center Co., Ltd.), NaCl (AR, Sinopharm Chemical Reagent Co., Ltd), KCl (99.90%, Greagent), $CaCl_2 \cdot 2H_2O$ (74%, Shanghai Silian Industry Co., Ltd.), $MgCl_2$ (AR, Adamas). All the chemicals were used without further purification.

### Synthesis of FJSM-CGTS

A mixture of $Cs_2CO_3$ (2.2 mmol, 0.7172 g), $Ga(NO_3)_3 \cdot xH_2O$ (0.7 mmol, 0.2566 g), Sn (0.7 mmol, 0.0842 g), S (6 mmol, 0.1927g), $N_2H_4 \cdot H_2O$ (98%, 1 mL) and $H_2O$ (0.5 mL) was heated in a 20 mL polytetrafluoroethylene (PTFE) lined stainless steel autoclave at 180 °C for 7 days and then the autoclave was brought to RT within 12 h. About 0.1656 g of pure light-pink plate-like crystal of FJSM-CGTS could be obtained with a yield of ~58.34% (based on Ga), which are stable in water and air. The atomic ratio of Ga/Sn in FJSM-CGTS is 1.40 by ICP-OES test.

### Synthesis of FJSM-KCGTS

Five hundred mg of FJSM-CGTS crystals were mixed with 500 mL solution of 2 mol $L^{-1}$ KCl and the mixture was shaken for 24 h at RT. Then the crystals were washed with deionized water and ethanol, and dried naturally to afford the FJSM-KCGTS.

### Synthesis of FJSM-KCGTS-Cs

Five hundred mg of FJSM-KCGTS crystals were mixed with 500 mL solution of 5000 mg $L^{-1}$ $Cs^+$ ions and shaken for 12 h at RT. Then the crystals were washed with deionized water and ethanol, and dried naturally to afford the FJSM-KCGTS-Cs.

### Characterizations

Single-crystal diffraction data for FJSM-CGTS and FJSM-KCGTS were collected with SuperNova CCD diffractometer with graphite monochromated Mo$K\alpha$ ($\lambda = 0.71073$ Å). Powder X-ray diffraction (PXRD) patterns were obtained at RT by using a Miniflex II diffractometer with Cu$K\alpha$ ($\lambda = 1.54178$ Å) at 30 kV and 15 mA in the angular range of $2\theta = 5$–$65°$. Energy dispersive spectroscopy (EDS), scanning electron microscope (SEM) and elemental distribution mapping analysis were

carried out through a JEOL JSM-6700F scanning electron microscope. X-ray photoelectron spectroscopy (XPS) analysis was carried out through a ESCALAB 250Xi spectrometer with Al$K\alpha$ radiation. Thermo Gravimetric Analysis (TGA) was performed on a NETZSCH STA 449F3 DTA–TG analyzer. Ion concentrations in solutions were measured by inductively coupled plasma-mass spectroscopy (ICP-MS) or inductively coupled plasma-optical emission spectroscopy (ICP-OES). ICP-MS and ICP-OES tests were carried out by XSerise II and Thermo 7400, respectively. FJSM-KCGTS crystal samples were irradiated with $\gamma$-rays at a total dose of 100 kGy (1.2 kGy $h^{-1}$ for 83.33 h) and 200 kGy (1.2 kGy $h^{-1}$ for 166.67 h) using a $^{60}Co$ irradiation source (2 million curies) provided by Detection Center of Suzhou CNNC Huadong Radiation Co., Ltd, China.

### Batch ion exchange experiments

In order to reduce the generation of radioactive waste during the experiments, non-radioactive $Cs^+$ ions were used to simulate $^{137}Cs^+$ ions for both batch ion exchange experiments and column experiments. A typical ion exchange experiment with FJSM-KCGTS was performed in a 20 mL polyethylene bottle containing an aqueous solution of cesium chloride ($V/m = 1000$ mL $g^{-1}$) and FJSM-KCGTS crystals. The polyethylene bottles containing the solid-liquid mixture were shaken in an oscillator at RT for 4 h and then left to stand for a few minutes. The supernatant was filtered through a filter with a 0.22 μm membrane and diluted to the appropriate concentration range for ICP testing. The removal rate ($R$) and distribution coefficient ($K_d$) could be calculated by Eqs. (1) and (2) (Supplementary Notes 1), respectively. The solid samples were washed several times with deionized water and anhydrous ethanol before drying.

In FJSM-CGTS activation kinetic experiments, 50 mg of FJSM-CGTS crystals were added to 50 mL of 2 mol $L^{-1}$ KCl solution under magnetic stirring at RT. The suspension was sampled at different time intervals and then filtered to test the concentration of $Cs^+$ in the solution. In the $Cs^+$ adsorption kinetic experiments by FJSM-KCGTS, 50 mg of FJSM-KCGTS crystals were added to 50 mL of 8.53 mg $L^{-1}$ CsCl solution under magnetic stirring at RT. The suspensions were sampled at different time intervals and then the solution samples were obtained by filtration to test the $Cs^+$ concentrations. The kinetic data of $Cs^+$ capture by FJSM-KCGTS were fitted with the pseudo-first-order kinetic model (Supplementary Notes 1, Eq. (3)) than the pseudo-second-order kinetic model (Supplementary Notes 1, Eq. (4)).

In isothermal experiments, $Cs^+$ solutions of different concentrations (0.982–854 mg $L^{-1}$ $Cs^+$, pH ~7) were prepared. The adsorption isotherm data of FJSM-KCGTS for $Cs^+$ at RT were fitted with the Langmuir isotherm model, Freundlich isotherm model and Langmuir-Freundlich isotherm models (Supplementary Notes 1, Eqs. (5)–(7)) to determine the $q_m$. In pH-dependent experiments, $Cs^+$ and $Sr^{2+}$ coexistence solutions were prepared at different pH (pH = 2.14–11.48), where the initial $Cs^+$ concentration was 4.87–5.05 mg $L^{-1}$ and $Sr^{2+}$ concentration was 4.95–5.05 mg $L^{-1}$. As well, $Cs^+$ isolated solutions were prepared at different pH (pH = 1.99–11.84), where the initial $Cs^+$ concentration was 4.92–5.07 mg $L^{-1}$. The acidity or alkalinity of the solutions was adjusted using $HNO_3$ or NaOH solutions. The leaching rate of metal ions was calculated by Eq. (8) (Supplementary Notes 1). In the competitive ion exchange experiments, $Na^+/Cs^+$, $Sr^{2+}/Cs^+$, $Mg^{2+}/Cs^+$, $Ca^{2+}/Cs^+$, and $K^+/Cs^+$ solutions were prepared. In the actual water samples, $Cs^+$ ions were added to river water (Fuzhou, Fujian), lake water (Fuzhou, Fujian) and seawater (Zhangzhou, Fujian) to simulate $Cs^+$ contaminated water bodies.

In the adsorption-desorption cycling experiment, 250 mg of FJSM-KCGTS-Cs was mixed with 250 mL of 1 mol $L^{-1}$ $NH_4Cl$ solution and shaken for 24 h to fully elute $Cs^+$ adsorbed by the material. Using the sample obtained at this time as the initial material for the cycling experiments ensures the consistency of the material used in each round of cycling. Each cycle contains two phases of adsorption and

desorption. The solution used in the adsorption process was 79.45 mg L$^{-1}$ CsCl solution with a contact time of 4 h. The solution used in the desorption process was 1 mol L$^{-1}$ NH$_4$Cl solution with a contact time of 12 h. After the completion of each adsorption/desorption, the supernatant was used to determine the Cs$^+$ concentration after filtration and dilution. The desorption rate could be calculated by Eqs. (9)–(11) (Supplementary Notes 1). The solid samples were washed and dried and then partially used for PXRD test to confirm the framework stability of the compounds and partially used for the next round of adsorption/desorption experiments. The solid samples used in each adsorption/desorption experiment were weighed and added to the corresponding amount of solution to ensure a solid-liquid ratio of 1000 mg L$^{-1}$.

### Actual $^{137}$Cs-liquid-waste treatment test
The thermal test was conducted using $^{137}$Cs-liquid-waste generated by Atomic High Tech of CNNC in industrial production. The initial activity concentrations of the generated neutral or acidic (pH = 2.6) $^{137}$Cs-liquid-wastes were $8.78 \times 10^4$ Bq mL$^{-1}$ and $1.53 \times 10^4$ Bq mL$^{-1}$, respectively, and the conductivities were 9.72 mS cm$^{-1}$ and 12.06 mS cm$^{-1}$, respectively. Five mg, 10 mg, and 40 mg of FJSM-KCGTS were added to 5 mL of neutral or acidic $^{137}$Cs waste solution, respectively. After shaking for 6 h, the supernatant was filtered and the activity of $^{137}$Cs was measured by a $\gamma$-hole counter.

### Column experiments
In total, 0.095 g of FJSM-KCGTS crystal sample was loaded into a polyethylene column with an inner diameter of 4.50 mm. The loading height was approximately 7.56 mm and the bed volume was 0.12 mL. A sieve plate with a pore size of 10 μm was placed at the bottom of the column to avoid loss of solid sample. A solution of 31.995 mg L$^{-1}$ Cs$^+$ was passed through the ion exchange column at a flow rate of 0.6 mL min$^{-1}$ (5 BV min$^{-1}$). The adsorption data were fitted with the Thomas model (Supplementary Notes 1, Eq. (12)). After adsorption was completed, the adsorbed Cs$^+$ was eluted using 1 mol L$^{-1}$ NH$_4$Cl solution as a drench solution. Samples of the effluent solution were collected in polyethylene tubes within each 5-min period and the measured concentration was approximated to the concentration at the intermediate moment. The Cs$^+$ ions concentration of the eluate collected from each tube was determined, and the total amount of Cs$^+$ eluted down could be calculated. The desorption rate could be calculated by Eq. (13) (Supplementary Notes 1). Solution flow rate and sample collection are controlled by a peristaltic pump and automatic collector.

### Density functional theory calculations
All calculations were performed using the Vienna Ab-initio Simulation Package (VASP, version number 6.2.1) with spin-polarized density functional theory (DFT) methods[47–49]. The generalized gradient approximation (GGA) with the Perdew-Burke-Ernzerhof (PBE) functional are used to describe the exchange and correlation energies[50,51]. The projector augmented-wave (PAW) method was used to describe the electron-ion interactions. Based on careful convergence tests, the plane wave energy cutoff was set to 400 eV. The convergence criterion of electronic structure was set to 10$^{-4}$ eV. The atomic relaxation was continued until the forces acting on atoms were smaller than 0.05 eV Å$^{-1}$. The Brillouin zone was sampled in a k-point mesh with the separation of 0.04 Å$^{-1}$. The Gaussian smearing of 0.05 eV was applied in order to speed up electronic convergence. In order to avoid the unwanted interaction between the slab and its period images, a vacuum height of 15 Å along the vertical direction was chosen. The adsorption binding energies ($E_{ads}$) used in this paper were calculated based on Eq. (14) (Supplementary Notes 1).

## Data availability
Crystallographic data for the structures reported in this Article have been deposited at the Cambridge Crystallographic Data Centre, under deposition numbers 2288484 (FJSM-CGTS), 2288487 (FJSM-KCGTS) and 2312868 (FJSM-KCGTS-Cs). Copies of the data can be obtained free of charge via https://www.ccdc.cam.ac.uk/structures/. The data that supports the findings of the study are included in the main text and Supplementary Information files. Raw data can be obtained from the corresponding author upon request. Source data are provided with this paper.

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

## Acknowledgements

We greatly thank financial support from National Natural Science Foundation of China and the Natural Science Foundation of Fujian Province. M.F. was supported by the National Natural Science Foundation of China (Nos. 22325605, U21A20296 and 22076185) and the Natural Science Foundation of Fujian Province (No. 2020J06033).

## Author contributions

The manuscript was written through contributions of all authors. J.T., M.F. and X.H. conceived the project. J.T. designed and performed most of the experiments and analyzed the data. S.J., J.L., L.Y. and H.S. offered help in the partial experiments. J.T. prepared the manuscript. M.F. and X.H. revised the manuscript. M.F. supervised the project.

## Competing interests

The authors declare no competing interests.
