## [Peer Review File · Nature Communications]

“Ion-imprinting” strategy towards metal sulfide scavenger enables the highly selective capture of radiocesiumREVIEWER COMMENTS

Reviewer #1 (Remarks to the Author):

The paper describes a new layered gallium thioostannate $\text{Cs}_{2.33}\text{Ga}_{2.33}\text{Sn}_{1.67}\text{S}_8\cdot\text{H}_2\text{O}$ (FJSM-CGTS) that exhibits the imprinting effect on Cs^+ . This is the first time that the ion imprinting method has been utilized for isolation of metal chalcogenides with selective ion sorption properties. Consequently, this research could pave the way for isolating several highly selective sorbents. The reported results for Cs capture demonstrate remarkable selectivity for this ion over several competitive ions. Furthermore, the sorbent displays remarkable efficiency in capturing Cs, even from real nuclear waste samples, underscoring its high potential for application in nuclear waste remediation. Overall, this work represents a significant contribution and is suitable for publication in Nature Communications. However, I have some comments/questions for the authors:

1. Lines 33,34: K^+ (212 ppm), Na^+ (228 ppm), Cs^+ (219 ppm) not ppm, but pm.
2. The authors mention (line 39, 40) that "However, to the best of our knowledge, there is no effective strategy or material for solving the problem that interfering ions in neutral solutions greatly affect the Cs^+ adsorption performance of ion exchangers".

The authors need to look at the review Chem. Sci., 2016, 7, 4804–4824 (reference 9 in the present paper). In page 4817 and Fig. 28, Cs^+ sorption data for the K6MS material performed at neutral solutions show exceptional selectivity of the metal sulfide for Cs^+ even in the presence of 10^4 or 10^5 molar excess of Na^+ or Ca^{2+} . Therefore, there is already a known material with exceptional selectivity for Cs^+ in the presence of interfering ions in a neutral solution. The selectivity of this sorbent is due to the pores of this material being suitable for Cs^+ accommodation. In addition, there is the layered $[\text{Ga}_2\text{Sb}_2\text{S}_7]^{2-}$ material with high selectivity for Cs^+ vs. Na^+ , K^+ and Sr^{2+} in neutral solutions, which is due to a Cs^+ capture mechanism reminiscent that of the Venus flytrap action (DOI: 10.1038/nchem.519).

3. In Fig. 1b: Cs_1 , Cs_2 , Cs_3 (CGTS) are replaced by K_1 , K_2 , K_3 (KCGTS) and still there is Cs. Is the Cs remaining after K^+ exchange in K/Cs mixed position?

4. Line 90, 91: "The close proximity between Cs_2 and Cs_2B leads to the inability of both positions to be occupied by Cs^+ at the same time, and similarly for Cs_3 and Cs_3B ".

What is the difference between Cs_2 and Cs_2B as well as Cs_3 and Cs_3B ? Is one pure Cs position and the other mixed Cs/K? Please explain.

5. How the experimental Cs^+ sorption capacity is compared with the theoretical based on the formula of the metal sulfide?

6. Line 120: "In addition, the Cs^+ adsorbed material can be regenerated by elution with 1 mol/L NH_4Cl solution, and the RCs can be maintained above 98.97% after three adsorption-elution cycles (Supplementary section 2.3)"... Why the authors have not used K^+ ions for regeneration?

7. The authors report that the Cs^+ sorption is energetically favored vs. K^+ , Na^+ , Eu^{3+} , Sr^{2+} .

What about the effect of pore size of the metal sulfide for the capture of Sr^{2+} and Eu^{3+} ? The hydrated radii of these cations are larger than those of Cs^+ and thus, their capture by the metal sulfide could be restricted by their larger sizes.

8. What is the role of hydrazine in the synthesis of the material?

Reviewer #2 (Remarks to the Author):

The selective removal of ^{137}Cs from complex radioactive wastewater remains an intractable issue related to the sustainable development of the nuclear industry and environmental safety. This manuscript proposes a special strategy for preparing inorganic adsorbents using the "ion-imprinting" method to enhance the selective adsorption of target ions. The prepared Cs^+ -imprinted adsorbent overcame the effect of excess competing ions to achieve selective capture of Cs^+ . In particular, it can effectively treat actual ^{137}Cs wastewater for waste minimization. The authors demonstrate the effectiveness of this synthesis strategy through exhaustive selective adsorption experiments. Moreover, the selective capture mechanism also clearly elucidated through powerful structural analysis and theoretical calculations. It is innovative that the proposed scheme allows the preparation of novel adsorbents combining advantages of high efficiency and stability of inorganic materials with the excellent selectivity of imprinted adsorbents. This work provides new possibilities for the design of novel inorganic ion exchangers with specific recognition-separation functions and is expected to contribute to the scientific understanding of structure-function relationships. The manuscript is well organized. Therefore, I suggest that this manuscript will be accepted after minor revision.

1. Some metal sulfides containing protonated amine ions have also gained Cs^+ removal properties by activation method. Do the properties of these materials are different from those reported in this paper? The authors should compare the selectivity of these compounds to further illustrate the effectiveness of the strategy of this paper.

2. What is the reason for the higher adsorption capacity obtained from column experiments than from adsorption isothermal experiments? The authors should explain this phenomenon.

3. Does the process of activation or exchange cause changes in the layer structure? The structural diagrams of the compounds FJSM-KCGTS as well as FJSM-KCGTS-Cs are missing in this manuscript.

4. Pictures of the experimental apparatus used for column separation experiments should be provided.

5. Some other errors that need to be fixed:

a. The ordering of the formulas in the Supplementary Information is wrong and the authors need to revise it. For example, Eq. 2 should be Eq. S2. The formula of elution rate and binding energy should be Eq. S13 and Eq. S14, respectively.

b. "T" of "T2 cluster" should be unified in italics. For example, lines 88 and 89.

Response to referees' comments

Reviewer: 1

The paper describes a new layered gallium thiostannate $\text{Cs}_{2.33}\text{Ga}_{2.33}\text{Sn}_{1.67}\text{S}_8 \cdot \text{H}_2\text{O}$ (FJSM-CGTS) that exhibits the imprinting effect on Cs^+ . This is the first time that the ion imprinting method has been utilized for isolation of metal chalcogenides with selective ion sorption properties. Consequently, this research could pave the way for isolating several highly selective sorbents. The reported results for Cs capture demonstrate remarkable selectivity for this ion over several competitive ions. Furthermore, the sorbent displays remarkable efficiency in capturing Cs, even from real nuclear waste samples, underscoring its high potential for application in nuclear waste remediation. Overall, this work represents a significant contribution and is suitable for publication in Nature Communications. However, I have some comments/questions for the authors.

Response: Thanks for the positive comments.

1. Lines 33,34: K^+ (212 ppm), Na^+ (228 ppm), Cs^+ (219 ppm) not ppm, but pm.

Response to the comment: We are sorry for this omission. These errors have been corrected.

2. The authors mention (line 39, 40) that “However, to the best of our knowledge, there is no effective strategy or material for solving the problem that interfering ions in neutral solutions greatly affect the Cs^+ adsorption performance of ion exchangers”.

The authors need to look at the review Chem. Sci., 2016, 7, 4804–4824 (reference 9 in the present paper). In page 4817 and Fig. 28, Cs^+ sorption data for the K_6MS material performed at neutral solutions show exceptional selectivity of the metal sulfide for Cs^+ even in the presence of 10^4 or 10^5 molar excess of Na^+ or Ca^{2+} . Therefore, there is already a known material with exceptional selectivity for Cs^+ removal in the presence of interfering ions in a neutral solution. The selectivity of this sorbent is due to the pores of this material being suitable for Cs^+ accommodation. In addition, there is the layered $[\text{Ga}_2\text{Sb}_2\text{S}_7]^{2-}$ material with high selectivity for Cs^+ vs. Na^+ , K^+ and Sr^{2+} in neutral solutions, which is due to a Cs^+ capture mechanism reminiscent that of the Venus flytrap action (DOI: 10.1038/nchem.519).

Response to the comment: Thanks for your suggestion and reminder. K_6MS material is undoubtedly a milestone in the use of metal sulfides for selective capture of Cs^+ and provide important reference for subsequent study. According to your suggestions, we have done further survey on the selective capture of Cs^+ . It can be seen that, in addition to K_6MS , there are still some three dimensional (3D) metal sulfides that exhibit good selectivity for Cs^+ removal (e.g., $[(\text{Me})_2\text{NH}_2]_{0.75}[\text{Ag}_{1.25}\text{SnSe}_3]$ and its isomorphous compounds AgSnSe-1 and CuGeS-1 , as reported in Dalton Trans., 2011, 40, 4387–4390; Chem. Commun., 2019, 55, 13884–13887; Inorg. Chem., 2020, 59, 9638–9647). Indeed, the pore size of these 3D materials plays an important role for the selective capture of Cs^+ . We also notice that the 2D layered $[\text{Ga}_2\text{Sb}_2\text{S}_7]^{2-}$ exhibits high selectivity for Cs^+ removal in neutral solutions due to a unique mechanism reminiscent that of the Venus flytrap action.

In order to compare the selectivity of the above materials and our title material under the same experimental conditions, the reported 3D- K_6MS , 3D- $[(\text{Me})_2\text{NH}_2]_{0.75}[\text{Ag}_{1.25}\text{SnSe}_3]$ and 2D- $[\text{Ga}_2\text{Sb}_2\text{S}_7]^{2-}$ have been synthesized as representatives of 3D or 2D materials and their Cs^+

selective capture experiments were carried out in seawater samples and solutions with different Sr/Cs or Eu/Cs molar ratios under neutral conditions. According to our experimental results and literature reports, suitable pore size is conducive to the selective capture of Cs⁺, but the pore size also limits the adsorption capacity and adsorption rate of these materials for Cs⁺ (Chem. Sci., 2016, 7, 4804–4824; Inorg. Chem., 2020, 59, 9638–9647).

In the experiments, FJSM-GAS-1 is chosen as the representative of [Ga₂Sb₂S₇]²⁻ material. The results of the selective experiment are as follows.

Supplementary Fig. 19. Comparison of Cs⁺ removal by 3D-K₆MS, 2D-FJSM-GAS-1, 3D-[(Me)₂NH₂]_{0.75}[Ag_{1.25}SnSe₃], and FJSM-KCGTS in (a) seawater samples, and in solutions with different (b) Sr/Cs and (c) Eu/Cs molar ratios.

The above results confirm that in seawater with high salinity and complex environment, 3D-[(Me)₂NH₂]_{0.75}[Ag_{1.25}SnSe₃] and 2D-FJSM-KCGTS can effectively capture Cs⁺, whereas the Cs⁺ capture performance of the 3D-K₆MS material and 2D-FJSM-GAS-1 will be impaired. In neutral solutions, although both 3D materials hardly adsorbed Sr²⁺ and Eu³⁺ ions, the presence of competing ions also resulted in a low removal rate of Cs⁺ even though the initial concentration of Cs⁺ was only about 5 mg/L. The presence of Sr²⁺ and Eu³⁺ also made the R^{Cs} of K₆MS be less than 60% and the R^{Cs} of [(Me)₂NH₂]_{0.75}[Ag_{1.25}SnSe₃] be less than 75% (Supplementary Fig. 18b-c). Under the same conditions, FJSM-KCGTS adsorbed Sr²⁺ and Eu³⁺, but it also maintained a high removal rate of Cs⁺. In Sr/Cs solution and Eu/Cs solution, the R^{Cs} values of FJSM-KCGTS can reach 98.98% and 98.90%, respectively. Even in the presence of high concentrations of competing ions, it can efficiently remove low concentrations of Cs⁺ in solution. The results confirm that our title material FJSM-KCGTS has better Cs⁺ selective capture performance compared with FJSM-GAS-1.

Overall, the selective Cs⁺ capture performance of FJSM-KCGTS synthesized under the guidance of our strategy is at the forefront of reported materials. Although 3D metal sulfide ion exchange materials traditionally have the selective Cs⁺ capture property owing to specific pore sizes, the limitation of pore sizes have also had a negative impact on the adsorption efficiency (in terms of removal rates, capacity, etc.) of 3D materials. Here, the synthetic strategy of “inorganic ion imprinted adsorbent” developed by us can enable the 2D materials to also possess the spatially limited domain effect, which makes the materials have both the advantages of high adsorption efficiency of 2D adsorbents and high selectivity of 3D adsorbents.

We have added a discussion of this issue in the Supplementary Information:

“2.5 Comparison with other materials”

Three-dimensional microporous metal sulfide ion exchangers with specific pore sizes have shown excellent Cs⁺ trapping properties. In order to compare the selectivity of FJSM-KGCTS with these

materials under the same experimental conditions, we synthesized 3D-K₆MS as well as 3D-[(Me)₂NH₂]_{0.75}[Ag_{1.25}SnSe₃] and performed their Cs⁺ selective capture experiments. In addition, we also synthesized FJSM-GAS-1 as a representative two-dimensional material. The synthesized samples were confirmed as pure phase by PXRD test (Supplementary Fig. 18).

The results shown in Supplementary Fig. 19a demonstrate that in seawater with high salinity and complex environment, 3D-[(Me)₂NH₂]_{0.75}[Ag_{1.25}SnSe₃] and 2D-FJSM-KCGTS can effectively capture Cs⁺, whereas the Cs⁺ capture performance of the 3D-K₆MS material and 2D-FJSM-GAS-1 is greatly affected. 3D-K₆MS and 3D-[(Me)₂NH₂]_{0.75}[Ag_{1.25}SnSe₃] hardly adsorb Sr²⁺ and Eu³⁺ ions in Sr/Cs or Eu/Cs solutions. However, the presence of competing ions also made the R^{Cs} values of 3D-K₆MS and 3D-[(Me)₂NH₂]_{0.75}[Ag_{1.25}SnSe₃] be less than 60% and 75%, respectively (Supplementary Fig. 19b-c), even though the initial concentration of Cs⁺ was only about 5 mg/L. By contrast, 2D-FJSM-KCGTS maintained a high removal rate of Cs⁺ (Supplementary Fig. 19b-c). Even in the presence of high concentrations of competing ions, it can efficiently remove low concentrations of Cs⁺ in solutions. In Sr/Cs and Eu/Cs solutions, the R^{Cs} values of 2D-FJSM-KCGTS can reach 98.98% and 98.90%, respectively. By contrast, the R^{Cs} range of FJSM-GAS-1 in Sr/Cs and Eu/Cs solutions is 7.89-51.82% and 1.37-34.57%, respectively, and the R^{Cs} is less than 1% in seawater samples. Therefore, compared with 2D-FJSM-GAS-1, 2D-FJSM-KCGTS has significantly better performance for selective capture of Cs⁺.

Supplementary Fig. 18. Experimental and simulated PXRD patterns of (a) 3D-K₆MS, (b) 2D-FJSM-GAS-1, and (c) 3D-[(Me)₂NH₂]_{0.75}[Ag_{1.25}SnSe₃].

Supplementary Fig. 19. Comparison of Cs⁺ removal by 3D-K₆MS, 2D-FJSM-GAS-1, 3D-[(Me)₂NH₂]_{0.75}[Ag_{1.25}SnSe₃], and FJSM-KCGTS in (a) seawater samples, and in solutions with different (b) Sr/Cs and (c) Eu/Cs molar ratios.

And we have added the relevant discussion in main text as follows:

“A very small number of three-dimensional materials with suitable pore size have good

selectivity for Cs⁺ capture^{9, 22-25}. However, we found that the presence of competing Sr²⁺ and Eu³⁺ ions is also detrimental to their capture efficiency for Cs⁺ (Supplementary section 2.5). The selective Cs⁺ capture performance of FJSM-KCGTS synthesized under the guidance of our strategy is at the forefront of reported materials, with high adsorption capacity, short adsorption equilibrium time and high selectivity. It suggests that the current “ion-imprinting effect” plays a key role in the Cs⁺ capture process of FJSM-KCGTS. The design and synthesis strategy of the current material should be effective in significantly enhancing the selectivity for Cs⁺ removal in neutral high-salinity solutions.”.

Based on the above results, we have modified the expression of “However, to the best of our knowledge, there is no effective strategy or material for solving the problem that interfering ions in neutral solutions greatly affect the Cs⁺ adsorption performance of ion exchangers” in the Introduction to the following one.

“Under neutral conditions, a very small number of three-dimensional microporous metal sulfide ion exchange materials with suitable pore sizes show good selectivity for Cs⁺ capture, but the pore size limitation also makes these materials inefficient for Cs⁺ removal, which is manifested by limited adsorption capacity and slow adsorption rate^{9, 22-25}. Although the adjustable interlayer spacing and readily exchangeable interlayer ions of layered sulfides allow them to trap Cs⁺ efficiently, their Cs⁺ trapping performance is greatly weakened by competing ions^{9, 26}. Therefore, it is still eager to develop the effective construction strategy for cesium scavengers to achieve efficient and highly selective capture of Cs⁺ ions.

”

3. In Fig. 1b: Cs1, Cs2, Cs3 (CGTS) are replaced by K1, K2, K3 (KCGTS) and still there is Cs. Is the Cs remaining after K⁺ exchange in K/Cs mixed position?

Response to the comment: In FJSM-KCGTS, the remaining Cs⁺ are in the mixed K/Cs position. This has been described in the Synthesis and Characterization section of the manuscript. “In FJSM-KCGTS, Cs⁺ in the original Cs1 and Cs3/Cs3B positions were completely replaced by K⁺ and Cs⁺ in the Cs2/Cs2B positions were partially replaced by K⁺.” We have added the structure diagrams of FJSM-KCGTS and FJSM-KCGTS-Cs to the Supplementary Information to make it easier for readers to understand our descriptions.

Supplementary Fig.7. View of layers stacking in (a) FJSM-KCGTS and (b) FJSM-KCGTS-Cs along the *a* axis. K3B, Cs3B, O, and H are ignored for clarity.

4. Line 90, 91: "The close proximity between Cs2 and Cs2B leads to the inability of both positions to be occupied by Cs⁺ at the same time, and similarly for Cs3 and Cs3B". What is the difference

between Cs2 and Cs2B as well as Cs3 and Cs3B? Is one pure Cs position and the other mixed Cs/K? Please explain.

Response to the comment: There are three crystallographically independent sites for Cs⁺ in FJSM-CGTS, namely Cs1, Cs2, and Cs3. Due to the disorder of Cs2 and Cs3, they were cleaved into two positions (Cs2/Cs2B and Cs3/Cs3B), respectively, during structure refinements. After exchanging K⁺ (FJSM-KCGTS), Cs⁺ on the Cs1 and Cs3/Cs3B sites were completely replaced by K⁺, and Cs⁺ on Cs2/Cs2B were partially replaced. Based on the consideration of the reasonableness of the Cs-S and K-S distances, the K⁺ is allowed to occupy the Cs2B position while the remaining Cs is occupying the Cs2 position during the structural refinement.

5. How the experimental Cs⁺ sorption capacity is compared with the theoretical based on the formula of the metal sulfide?

Response to the comment: The theoretical exchange capacity of FJSM-KCGTS for Cs⁺ was 287.52 mg/g, calculated from the chemical formulae of FJSM-KCGTS and FJSM-KCGTS-Cs. The experimental values (246.65 mg/g) were slightly lower than the theoretical values which may be attributed to the fact that the exchanged K⁺ would compete with Cs⁺. The crystals of FJSM-KCGTS-Cs were obtained in a 5000 mg/L Cs⁺ solution. Since the ICP-MS test requires a solution concentration of less than 100 µg/L, solutions with high Cs⁺ concentration were not used in the isothermal adsorption experiments in order to avoid large testing errors due to excessive dilution.

We have added a discussion of this issue in the main text:

“The theoretical exchange capacity of FJSM-KCGTS for Cs⁺ was 287.52 mg/g, calculated from the chemical formulae of FJSM-KCGTS and FJSM-KCGTS-Cs. The experimental values were slightly lower than the theoretical value which may be attributed to the fact that the exchanged K⁺ would compete with Cs⁺.”

6. Line 120: "In addition, the Cs⁺ adsorbed material can be regenerated by elution with 1 mol/L NH₄Cl solution, and the RCs can be maintained above 98.97% after three adsorption-elution cycles (Supplementary section 2.3)"... Why the authors have not used K⁺ ions for regeneration?

Response to the comment: We hoped that the elution process would not only achieve material regeneration, but also enable the eluted Cs⁺ to be recycled again. Therefore, we did not use K⁺ to elute the material after adsorption of Cs⁺ to avoid the excessive K⁺ to go back to the elution solution which is difficult to be further processed. As for the NH₄⁺ ions in the solution, there are mature treatment methods available.

7. The authors report that the Cs⁺ sorption is energetically favored vs. K⁺, Na⁺, Eu³⁺, Sr²⁺. What about the effect of pore size of the metal sulfide for the capture of Sr²⁺ and Eu³⁺? The hydrated radii of these cations are larger than those of Cs⁺ and thus, their capture by the metal sulfide could be restricted by their larger sizes.

Response to the comment: As we have discussed in the response to the second comment, the pore size of three-dimensional metal sulfides has a large impact on their ion exchange properties and will limit their ability to capture larger sized cations. By contrast,

two-dimensional materials have a more flexible structure compared to three-dimensional materials, and the cations between the layers are more easily exchanged, making it difficult to sieve cations through size limitations. For example, KMS-1, KMS-2, KTS-3, and FJSM-SnS materials with plate-layer stacking mode have adsorption properties for Cs^+ , Sr^{2+} , and other metal cations (Chem. Sci. 7, 4804-4824). The characteristic of flexible and adjustable interlayer spacing for these 2D ion exchangers makes them easily accommodate foreign ions, which may lead to their relatively poor selective capture of Cs^+ . In the selectivity experiments in this manuscript, it can be found that FJSM-KCGTS adsorbs Cs^+ along with Sr^{2+} and Eu^{3+} due to the redundancy of adsorption sites. However, it is distinctive that in the presence of excess competing ions, FJSM-KCGTS can still effectively capture low concentrations of Cs^+ in solution. When the content of Cs^+ is comparable to that of Sr^{2+} , FJSM-KCGTS is more inclined to capture Cs^+ . Overall, FJSM-KCGTS shows more outstanding Cs^+ selectivity than that of the Cs^+ ion exchangers with plate-layer stacking mode. We believe that the wavy layers stacking pattern of FJSM-KCGTS produces stronger interlayer interactions than planar layers stacking pattern, favoring the reduction of layer spacing variations, and thus provides a suitable space and coordination environment for Cs^+ . In other words, the wavy layered structure and the strong interaction of soft basic S^{2-} sites with Cs^+ provide a spatially confined effect on Cs^+ , contributing to the excellent selectivity of FJSM-KCGTS for Cs^+ capture. This production of suitable sites and coordination environments for Cs^+ is closely related to the method of “ion imprinting”. Combined with the results of DFT calculations, we believe that “ion imprinting” makes the adsorption of Cs^+ by FJSM-KCGTS be more energetically favourable than that of K^+ , Na^+ , Sr^{2+} , and Eu^{3+} .

8. What is the role of hydrazine in the synthesis of the material?

Response to the comment: We believe that hydrazine hydrate plays a role in the synthesis by providing a strong basic environment and promoting crystallisation. The amount of hydrazine hydrate added will have an effect on the yield and crystallinity of the product. The role of hydrazine hydrate as a co-solvent and facilitator of crystallization has also been mentioned in some previous literatures (Angew. Chem. Int. Ed. 2011, 50, 11395–11399; Inorg. Chem. 2015, 54, 5874–5878).

Reviewer: 2

The selective removal of ^{137}Cs from complex radioactive wastewater remains an intractable issue related to the sustainable development of the nuclear industry and environmental safety. This manuscript proposes a special strategy for preparing inorganic adsorbents using the "ion-imprinting" method to enhance the selective adsorption of target ions. The prepared Cs^+ -imprinted adsorbent overcame the effect of excess competing ions to achieve selective capture of Cs^+ . In particular, it can effectively treat actual ^{137}Cs wastewater for waste minimization. The authors demonstrate the effectiveness of this synthesis strategy through exhaustive selective adsorption experiments. Moreover, the selective capture mechanism also clearly elucidated through powerful structural analysis and theoretical calculations. It is innovative that the proposed scheme allows the preparation of novel adsorbents combining advantages of high efficiency and stability of inorganic materials with the excellent selectivity of imprinted adsorbents. This work provides new possibilities for the design of novel inorganic ion exchangers with specific recognition-separation functions and is expected to contribute to the scientific understanding of structure-function relationships. The manuscript is well organized. Therefore, I suggest that this manuscript will be accepted after minor revision.

Response: Thanks for the positive comments.

1. Some metal sulfides containing protonated amine ions have also gained Cs^+ removal properties by activation method. Do the properties of these materials are different from those reported in this paper? The authors should compare the selectivity of these compounds to further illustrate the effectiveness of the strategy of this paper.

Response to the comment: Thanks for the suggestion. Some metal sulfides containing protonated organic amine cations have been activated for adsorption of Cs^+ , UO_2^{2+} ions (Inorg. Chem. 2019, 58, 11622–11629; Environ. Sci.: Adv., 2022, 1,331–341; Inorg. Chem. 2023, 62, 12843–12850). Neutral sulfides (MPS_3) have also been used to remove Cs^+ and UO_2^{2+} after intercalation activation (Chem. Eur. J. 2017, 23, 11085–11092; Chem. Eng. J. 2022, 429, 132474). Obviously, the activation method is an effective way to enhance the adsorption performance for some metal sulfides. Unfortunately, the study of this group of materials is still in its infancy, and their adsorption behavior in the coexistence of high-valent competing ions (especially Sr^{2+} and Eu^{3+}) with Cs^+ has not been systematically investigated. Therefore, a systematic comparison is not possible. However, from the available results, it was found that 3D-K@GaSnS-1 ($\text{K}_2\text{Ga}_{2.2}\text{Sn}_{1.8}\text{S}_{7.9}\cdot 9\text{H}_2\text{O}$) showed a significant decrease in Cs^+ removal performance in the presence of high concentrations of Na^+ , K^+ , Ca^{2+} , and Mg^{2+} . 2D-KIAS ($\text{K}_2\text{In}_2\text{Sb}_2\text{S}_{7.2}\cdot 2\text{H}_2\text{O}$) showed a significant decrease in Cs^+ removal performance in the presence of high concentrations of Na^+ , and Sr^{2+} . The above facts show that K^+ activation does not guarantee excellent selectivity of the material towards Cs^+ . The excellent selectivity of FJSM-KCGTS for Cs^+ capture stems from the relatively robust wavy layer structure and the strong interaction of soft basic S^{2-} sites with Cs^+ which provide a spatially confined effect on Cs^+ . That is, the current "ion imprinting effect" plays a key role in the Cs^+ capture process of FJSM-KCGTS.

We have added the discussion in the main text:

“Some metal sulfide ion-exchange materials containing protonated organic amines can also

capture Cs^+ after K^+ activation, but their trapping performance is also very susceptible to competing ions^{40, 42-43}...The selective Cs^+ capture performance of FJSM-KCGTS synthesized under the guidance of our strategy is at the forefront of reported materials, with high adsorption capacity, short adsorption equilibrium time, and high selectivity. It suggests that the current “ion-imprinting effect” plays a key role in the Cs^+ capture process of FJSM-KCGTS. The design and synthesis strategy of the current material should be effective in significantly enhancing the selectivity for Cs^+ removal in neutral high-salinity solutions.”

2.What is the reason for the higher adsorption capacity obtained from column experiments than from adsorption isothermal experiments? The authors should explain this phenomenon.

Response to the comment: Thanks for the suggestion. In adsorption isothermal experiments, the exchanged K^+ is present in the solution and will compete with Cs^+ for adsorption, whereas in the column experiment, the exchanged K^+ will be eluted out of the adsorption column and will not affect the adsorption of Cs^+ . Therefore, the adsorption capacity obtained from the adsorption isothermal experiment (246.65 mg/g) is slightly lower than the theoretical value (287.52 mg/g), while the adsorption amount obtained from the column experiment (288.94 mg/g) is comparable to the theoretical value.

We have added a discussion of this issue in the main text:

“Note that unlike isothermal adsorption experiments, the exchanged K^+ ions are eluted out of the adsorption column in the column experiments and do not affect the adsorption of Cs^+ . Therefore, the dynamic adsorption capacity is comparable to the theoretical value (287.52 mg/g).”

3.Does the process of activation or exchange cause changes in the layer structure? The structural diagrams of the compounds FJSM-KCGTS as well as FJSM-KCGTS-Cs are missing in this manuscript.

Response to the comment: Thanks for the suggestion. Single-crystal structural analyses showed that the cations between the interlayers are exchanged during the activation and exchange process, and the anionic layered structure and its stacking mode of the material does not change. We have added structural diagrams of FJSM-KCGTS and FJSM-KCGTS-Cs to the Supplementary Information.

Supplementary Fig.7. View of layers stacking in (a) FJSM-KCGTS and (b) FJSM-KCGTS-Cs along the *a* axis. K3B, Cs3B, O, and H are ignored for clarity.

4.Pictures of the experimental apparatus used for column separation experiments should be

provided.

Response to the comment: Thanks for the suggestion. We have added a diagram of the experimental setup to the Supplementary Information.

Supplementary Fig. 21. Experimental setup for ion exchange column experiments.

5. Some other errors that need to be fixed:

a. The ordering of the formulas in the Supplementary Information is wrong and the authors need to revise it. For example, Eq. 2 should be Eq. S2. The formula of elution rate and binding energy should be Eq. S13 and Eq. S14, respectively.

b. “T” of “T2 cluster” should be unified in italics. For example, lines 88 and 89.

Response to the comment: We are sorry for these omissions. These errors have been corrected.

REVIEWERS' COMMENTS

Reviewer #1 (Remarks to the Author):

The authors have addressed the reviewers' suggestions and further improved their paper. Thus, I am pleased to recommend accepting the paper for publication in Nature Communications.

Reviewer #2 (Remarks to the Author):

I'm satisfied for the revision and strongly recommend it publishing in Nat. Commun.

Response to referees' comments

Reviewer: 1

Comments:

The authors have addressed the reviewers' suggestions and further improved their paper. Thus, I am pleased to recommend accepting the paper for publication in *Nature Communications*.

Response: Thanks for the positive comments.

Reviewer: 2

Comments:

I'm satisfied for the revision and strongly recommend it publishing in *Nat. Commun.*.

Response: Thanks for the positive comments.